Journal of
**open** psychology data

**DATA PAPER**

]u[ ubiquity press

# Data from "A Registered Report Testing the Effect of Sleep on DRM False Memory: Greater Lure and Veridical Recall but Fewer Intrusions After Sleep"

MATTHEW H. C. MAK ⓘD

## ABSTRACT

This paper describes a rich dataset from a registered report investigating sleep's effect on false memory in the Deese-Roediger-McDermott (DRM) paradigm. 534 young adults completed free recall either shortly or 12 hours after studying lists of semantic associates (e.g., *hospital, nurse*). Collected online, our recall data showcase high data quality, replicating classic behavioural effects (e.g., serial position curve). The dataset contains raw recall data with original spelling and recall order, accompanied by demographic information (e.g., gender, time-of-day preference). Its versatility supports reuse in modelling memory decay and search processes, understanding lexical effects and individual differences, and benchmarking online memory studies.

**CORRESPONDING AUTHOR:**
**Matthew H. C. Mak**
University of Warwick, United Kingdom
Matthew.Mak@warwick.ac.uk

**KEYWORDS:**
False Memory; DRM; Recall; Sleep

**TO CITE THIS ARTICLE:**
Mak, M. H. C. (2024). Data from "A Registered Report Testing the Effect of Sleep on DRM False Memory: Greater Lure and Veridical Recall but Fewer Intrusions After Sleep". *Journal of Open Psychology Data,* 12: 6, pp. 1–15. DOI: https://doi.org/10.5334/jopd.98

# 1. BACKGROUND

## 1.1 SLEEP AND DRM FALSE MEMORY

Newly acquired declarative memories tend to be better remembered after a period of sleep compared to a similar duration of wakefulness (e.g., Ashton & Cairney, 2021; Mak et al., 2024; Plihal & Born, 1997; Potkin & Bunney, 2012). Some theories attribute this benefit to sleep-related consolidation, during which hippocampal replay reactivate the newly acquired memory, facilitating its assimilation into long-term neocortical stores (e.g., Klinzing et al., 2019; Paller et al., 2021; Stickgold, 2005). Alternatively, sleep may protect newly encoded memories from external interference, resulting in less forgetting (Jenkins & Dallenbach, 1924; Yonelinas et al., 2019).

In recent years, mounting evidence has suggested that the role of sleep extends beyond what was previously encoded (Horváth et al., 2016; Lewis & Durrant, 2011; Lutz et al., 2017). One strand of research in this area is how sleep impacts false memories, where individuals recall events or items that never actually happened (Calvillo et al., 2016; Darsaud et al., 2011; van Rijn et al., 2017). The Deese-Roediger-McDermott (DRM) paradigm is perhaps the most widely used paradigm for inducing false memories in the laboratory (Deese, 1959; Roediger & McDermott, 1995). Here, participants study lists of semantic associates ("studied list words" like *nurse, hospital, sick*), but not a "critical lure" that represents the gist of the list (e.g., *doctor*). In later memory tests (typically free recall or recognition), participants often mistakenly recall the critical lures or identify them as previously seen, despite not encountering them before. This DRM false memory effect has been widely studied and replicated across age groups (e.g., Sugrue & Hayne, 2006), speakers of different languages (e.g., Bialystok et al., 2020), presentation modalities (e.g., Cleary & Greene, 2002; Smith & Hunt, 1998), and various delay intervals between wordlist presentation and the subsequent memory test (e.g., from a few minutes to 60 days later; Seamon et al., 2002). Among these, a number of studies have examined whether a period of sleep, as opposed to wakefulness, during the delay interval affects the incidence of DRM false memory.

## 1.2 THEORETICAL SIGNIFICANCE

Understanding how sleep (vs. wake) may influence DRM false memory has substantial theoretical value, because vastly different predictions have been made by existing memory/sleep models. For example, the Information Overlap to Abstract model (iOtA; Lewis & Durrant, 2011) postulates that sleep plays a crucial role in gist abstraction, leading to the explicit prediction that sleep (vs. wake) would increase false memory for the critical lures (which represent the gist of each list). Conversely, an alternative proposition (e.g., Fenn et al., 2009; Lo et al., 2014) posits that sleep could potentially reduce DRM false memory. Specifically, it argues that sleep

may facilitate the consolidation of studied list words, which may, in turn, aid the suppression of critical lures, potentially via some kind of source monitoring processes (e.g., Kensinger & Schacter, 1999).

Our study, in addition to disentangling those opposing theories, serves to refine existing theoretical frameworks, including Fuzzy-Trace Theory (Brainerd & Reyna, 1998) and the Activation/Monitoring Framework (Roediger et al., 2001). These two theories have dominated the DRM literature (see Chang & Brainerd, 2021 for a review), but they are not sleep-specific theories, and are therefore mute on the role of sleep. Therefore, understanding how sleep may influence DRM false memory will help tighten these theories.

## 1.3 WEAK AND CONTRADICTORY EVIDENCE BASE

To-date, about 10 published studies have investigated whether a delay interval containing sleep (vs. wakefulness) affects the incidence of DRM false memories (e.g., Fenn et al., 2009; Payne et al., 2009; McKeon et al., 2012). Despite using similar experimental design and materials, these studies have yielded conflicting results: Some showed a post-sleep increase in DRM false memory (McKeon et al., 2012; Payne et al., 2009; Shaw & Monaghan, 2017), others reported no overall effect (Diekelmann et al., 2010), and some reported a reduction in false memories following sleep (Fenn et al., 2009; Lo et al., 2014). In short, the evidence base on how sleep may influence DRM false memory remains weak and inconsistent. In light of this, Newbury and Monaghan (2019) conducted a meta-analysis of 12 'Sleep × DRM' experiments. They reported that while sleep did not have a consistent effect, the effect of sleep was moderated by a key factor: the number of semantic associates in a DRM list. Specifically, a consistent post-sleep increase in false memories was observed when each list contains fewer associates (N = 10 words/list), whereas no consistent sleep effect emerged when each list contained more (N = 12 or 15 words/list). Newbury and Monaghan's (2019) meta-analysis pointed us towards the most conducive parameter for detecting a sleep effect in the DRM paradigm (i.e., short list length); however, the literature lacks a well-powered empirical study that tests the validity of this parameter. This is the backdrop against which our registered report takes a centre stage, where we used exclusively short DRM lists to investigate the effect of sleep.

## 1.4 OUR REGISTERED REPORT

We conducted a 2 (Interval) × 2 (Test Time) between-participant DRM experiment, which used free recall as the sole outcome measure. The first independent variable, Interval, refers to whether participants completed free recall either shortly or 12 hours after studying the DRM wordlists. The second independent variable, Test Time, refers to whether participants completed free recall either in the morning or in the evening. Our experimental

Mak *Journal of Open Psychology Data* DOI: 10.5334/jopd.98

design, therefore, resulted in four groups: Immediate-AM, Immediate-PM, Delay-AM (Sleep), and Delay-Wake (Wake). The inclusion of the Immediate groups helped rule out potential circadian effects on encoding/retrieval.

Our sample size was 488 young adults (after exclusion), giving us over 90% statistical power (alpha = 0.05) to detect our desired sleep effects. Our dataset, to the best of our knowledge, stands as the largest publicly accessible DRM free recall data. Regardless of whether researchers are interested in the effect of sleep, they may use our recall data for (1) modelling time-related memory decay and memory search processes, (2) exploring the relation between lexical properties and recall rates, (3) investigating how individual differences in e.g., veridical recall, contribute to false recall, and (4) using our data to benchmark memory studies conducted online.

## 2. METHODS

### 2.1 STUDY DESIGN

Our experiment had a 2 (Interval: Immediate vs. Delay) × 2 (Test Time: AM vs. PM) fully between-participant design. Figure 1 summarises the experimental procedure.

### 2.2 TIME OF DATA COLLECTION

Data collection took place between 01 December 2022 and 02 April 2023.

### 2.3 LOCATION OF DATA COLLECTION

Participants were recruited online via Prolific (https://www.prolific.co/). All participants completed the experiment unsupervised and at a location of their own choosing. Access to the experiment was restricted solely to individuals possessing a British IP address, as we need to ensure participation at a certain time of day.

### 2.4 SAMPLING, SAMPLE AND DATA COLLECTION

Following our prior sleep studies conducted via Prolific (Ball et al., 2024; Mak et al., 2023; 2024; Mak & Gaskell, 2023; Experiment 2), an 'expression of interest' survey (available in Appendix A) was administered on Qualtrics to recruit a pool of participants (*N* = 2296). The first part of the survey collected essential demographic details: gender identity, age, current country of residence, primary language, ethnicity, highest education level, and history of developmental/sleep disorders (if applicable). The second part of the survey outlined the main study, explaining that participants would be randomly allocated to one of the four groups and that no preferences would be accommodated. Participants indicated whether or not they would like to take part in the main study. Of the 2296 respondents, 1940 expressed interest in taking part, who were then screened for their eligibility:

1. Aged 18–25
2. Speaks English as (one of) their first language(s)
3. No known history of any psychiatric (e.g., schizophrenia), developmental (e.g., dyslexia) or sleep (e.g., insomnia) disorders
4. Currently resides in the UK, indexed by their IP address
5. Normal vision or corrected-to-normal vision
6. Normal hearing
7. Able to complete the study using a laptop or a desktop PC
8. Able to complete both the study and test phases
9. Has an approval rate of >96% on Prolific (This helps ensure that a participant tends to take online studies seriously).

Those who fitted the above inclusion criteria (*N* = 904) were randomly allocated to one of the four experimental

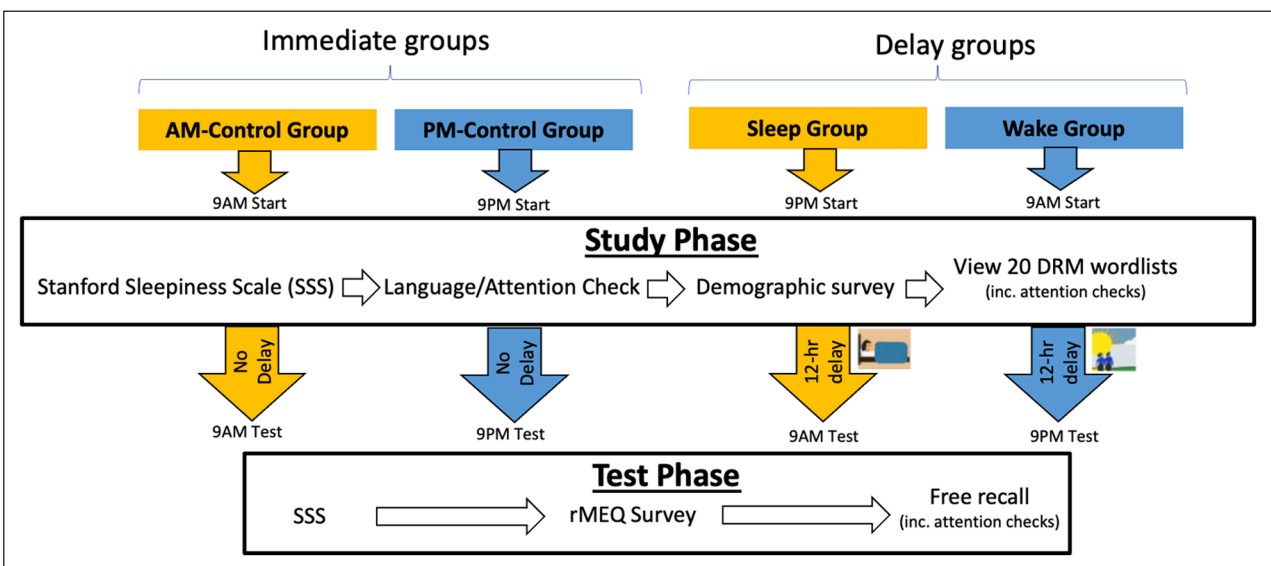

**Figure 1** Experimental procedure.

groups. In the end, 534 participants completed *both* the study and test phases. All participants were reimbursed at a rate of £9.5/hr, but those in the Delay groups received a bonus of £0.2 after completion of both sessions. Of the 534 participants, 46 were excluded from our confirmatory analyses for meeting one or more of our pre-registered exclusion criteria:

1. (Sleep and Wake groups only) they reported consuming any alcoholic drinks between study and test. ($N = 11$)
2. (Sleep group only) they reported to have had fewer than 6 hours of overnight sleep prior to test or rated their sleep quality as poor or extremely poor. ($N = 5$)
3. (Wake group only) they reported to have had a nap between study and test. ($N = 11$)
4. they failed more than one of the four attention checks (i.e., 3 + 3 = 11) at study. ($N = 6$)
5. they failed to report more than one of the four digits in the attention check of free recall. ($N = 12$)
5. they submitted a blank response in free recall. ($N = 1$)

Our confirmatory analyses encompassed a total of 488 participants, with 124 in each of the Immediate groups, and 120 in each of the Delay groups (see Table 1 for group characteristics). Note that although 46 participants were excluded from our analyses, their recall and survey data are available in the published datasets.

## 2.5 MATERIALS/SURVEY INSTRUMENTS

We made use of 20 DRM wordlists from Stadler et al. (1999), who normed a total 36 lists. The 20 we chose elicited the highest false recall rates in their study. We used the first 8 associates of each list, and as per the standard DRM paradigm, they were arranged in a descending order of associative strength to the critical lures (see Table 2). A participant studied all 20 lists, and therefore, a total of 160 list items.

We note that the original DRM lists by Stadler et al. (1999) were tailored for American participants, and two words (e.g., trash, Mississippi) were not immediately relatable to people in the UK. We, therefore, changed these words (e.g., *trash → rubbish*), as noted in Table 2.

### 2.5.1 Study phase

Participants first gave informed consent, rated their level of sleepiness on the Stanford Sleepiness Scale (SSS; Hoddes et al., 1973), completed a language/attention check (detailed in the next section), filled in a demographic survey, and studied 20 DRM wordlists. At wordlist exposure, participants were instructed that they would view some English words on a computer screen and were urged to attentively study them. Participants were told that their "memory for the words will be tested later", but there was no mention of the specific test format (i.e., participants were not told that they would engage in free recall).

During wordlist presentation, words within each DRM list were visually displayed in a fixed order, as in previous DRM studies (see Table 2). Each list initiated with a 1-second fixation, succeeded by the first word displayed for 1 second in lowercase black font (Arial, size 26) against a white backdrop. A 500-millisecond interstimulus gap separated each word. Following the final word in a list, a blank screen was shown for 5 seconds before the next list began. List order was randomised, and each list was shown once (hence 160 studied words in total). An attention check was integrated into the study task, outlined in Section 2.6.1 below.

| CHARACTERISTICS | IMMEDIATE-AM | IMMEDIATE-PM | SLEEP (AKA DELAY-AM) | WAKE (AKA DELAY-PM) |
|---|---|---|---|---|
| N before exclusion | 130 | 127 | 134 | 143 |
| N after exclusion | 124 | 124 | 120 | 120 |
| Mean age (SD) | 22.24 (2.18) | 22.34 (2.03) | 22.18 (1.93) | 22.25 (1.93) |
| Gender (Female: Male: Other) | 68 : 54 : 2 | 77 : 46 : 1 | 62 : 57 : 1 | 58 : 61 : 1 |
| Ethnicity (Asian: Black/Caribbean: Latino: Mixed: Other: White) | 11 : 6 : 0 : 9 : 1 : 97 | 21: 4: 0: 7: 1: 91 | 9 : 7 : 0 : 6 : 0 : 98 | 13: 4: 1: 4: 2: 96 |
| Education Attainment (A Level: Degree: GCSE: Postgrad: Vocational) | 47: 44: 6: 25: 2 | 54: 48: 3: 16: 3 | 58: 45: 3: 13: 1 | 51: 45: 2: 20: 2 |
| Mean SSS rating at study (SD) | 2.58 (0.98) | 2.64 (1.12) | 2.66 (0.96) | 2.58 (0.98) |
| Mean SSS rating at test (SD) | 2.73 (1.04) | 2.95 (1.29) | 2.63 (1.21) | 2.67 (1.18) |
| Mean rMEQ score (SD) | 15.89 (1.67) | 15.59 (1.91) | 15.72 (1.83) | 15.53 (1.99) |
| Mean N of intervening hr between study & test (SD) | NA | NA | 12.22 (0.74) | 12.14 (0.81) |

**Table 1** Group characteristics.

*Notes.* (1) SSS stands for Stanford Sleepiness Scale (Hoddes et al., 1973) and ranges from 1 to 6, with higher values indicating greater sleepiness. (2) rMEQ stands for reduced Morningness/Eveningness Questionnaire (Adan & Almirall, 1991); it ranges from 5 to 25, with higher values indicating greater preference for morning.

| CRITICAL LURE OF EACH LIST | FALSE RECALL PROBABILITY (STADLER ET AL., 1999) | LIST ITEMS (ARRANGED IN THE ORDER OF PRESENTATION IN STUDY) |
|---|---|---|
| Window | 65 | door, glass, pane, shade, ledge, sill, house, open |
| Sleep | 61 | bed, rest, awake, tired, dream, wake, snooze, blanket |
| Doctor | 60 | nurse, sick, lawyer, medicine, health, hospital, dentist, physician |
| Smell | 60 | nose, breathe, sniff, aroma, hear, see, nostril, whiff |
| Chair | 54 | table, sit, legs, seat, couch, desk, recliner, sofa |
| Smoke | 54 | cigarette, puff, blaze, billows, pollution, ashes, cigar, chimney |
| Sweet | 54 | sour, candy, sugar, bitter, good, taste, tooth, nice |
| Rough | 53 | smooth, bumpy, road, tough, sandpaper, jagged, ready, coarse |
| Needle | 52 | thread, pin, eye, sewing, sharp, point, prick, thimble |
| Rubbish (**Note 1**) | 49 | garbage, waste, can, refuse, sewage, bag, junk, trash (**Note 1**) |
| Anger | 49 | mad, fear, hate, rage, temper, fury, ire, wrath |
| Soft | 46 | hard, light, pillow, plush, loud, cotton, fur, touch |
| City | 46 | town, crowded, state, capital, streets, subway, country, New York |
| Cup | 45 | mug, saucer, tea, measuring, coaster, lid, handle, coffee |
| Cold | 44 | hot, snow, warm, winter, ice, wet, frigid, chilly |
| Mountain | 42 | hill, valley, climb, summit, top, molehill, peak, plain |
| Slow | 42 | fast, lethargic, stop, listless, snail, cautious, delay, traffic |
| River | 42 | water, stream, lake, Thames (**Note 2**), boat, tide, swim, flow |
| Spider | 37 | web, insect, bug, fright, fly, arachnid, crawl, tarantula |
| Foot | 35 | shoe, hand, toe, kick, sandals, soccer, yard, walk |

**Table 2** The 20 DRM wordlists used in the experiment.

*Note 1.* In Roediger et al. (2001), the critical lure for this list was *trash,* with *rubbish* being one of the list items. We used *rubbish* as the critical lure and *trash* as a list item because the former is the preferred term in British English.

*Note 2.* The original word in Roediger et al. was *Mississippi*. We replaced it with *Thames*.

For participants in the sleep and wake groups, Session 1 ended after wordlist exposure. On the final page, they were told that they should take part in the second session, which would be available 12 hours later, accessible via Prolific. They were also encouraged to put down an email address if they wished to be reminded via email. We programmed the experiment such that participants who put down an email address received an automated email when the second session became available on Prolific. Over 75% of the participants chose to supply an email address.

### 2.5.2 Test phase
Participants began by giving an SSS rating and completing a reduced morningness/evening Questionnaire (rMEQ; Adan & Almirall, 1991), which also asked participants to indicate how bright/noisy their immediate surrounding is, their sleep duration/quality the night before[1] and whether they had a period of nap between sessions (Wake group only).[2]

Finally, the test phase concluded with a 10-minute free recall task where participants recalled as many of the

words as they could from the previously seen wordlists. Participants typed their responses into a designated text box. A timer appeared when there were 2 minutes remaining, and participants could not proceed before the time was up. There was an attention check incorporated in this task, detailed in Section 2.6.1.

### 2.5.3 Data pre-processing
The free recall data were pre-processed ahead of data analysis. The first step was to remove any duplicate responses. The second was to correct all obvious spelling and typing errors to the nearest English words, defined as Levenshtein distance ≤ 2 (e.g., *cigerette → cigarette*). Responses with added or dropped inflectional suffixes (i.e., -s, -ed, -ing, adjectival -er) were corrected. Responses with derivational changes were considered as intrusions. For instance, one of the studied words was *pollution*; if a participant recalled *pollutions*, the plural suffix was dropped; however, if a participant recalled *pollutant*, this was considered as an intrusion (i.e., neither a studied list word nor a critical lure).

## 2.6 QUALITY CONTROL

To ensure the best possible data quality, we followed our prior online studies (e.g., Curtis et al., 2022; Mak, 2021) by incorporating several attention checks into the experiment, detailed below.

### 2.6.1 Attention checks

At the beginning of the study, participants were asked to do their best and were told that their participation was important for science. Afterwards, participants were told to turn on their audio so they could listen to a short story in English (14 sec). Participants heard the story once, without the option to replay or pause. Participants then responded to two simple comprehension questions. These served as language/attention checks, ensuring that participants understood English and were in a reasonably quiet place. Three individuals did not get both questions right, so the study terminated there for these individuals. Since they could not complete the study/test phase, they did not meet our inclusion criteria (and therefore not counted towards the sample size).

During wordlist presentation, there were a series of surprise attention checks. After the 4th, 9th, 13th, 18th lists, participants saw an erroneous maths equation such as "3 + 3 = 11". It was presented for 1 s, in the same font and style as the list words. Immediately afterwards, participants were asked to report what 3 + 3 was according to what was just shown. Participants were excluded from further analyses if they failed more than one of attention checks.

At free recall, to maximise the likelihood that participants paid full attention instead of doing something else (e.g., using their phone), there was an attention check throughout: Below the recall textbox, there was a white square that turned red every 2 to 3 minutes. This colour change persisted for 10 seconds, revealing a single digit. Participants were required to input this digit into a separate textbox, indicating their attentiveness. Over the course of the 10-minute recall task, the square turned red on four occasions, necessitating the input of four digits as participants engaged in recall. Participants were excluded from further analyses if they failed to report more than one digit.

Lastly, in order to discourage participants from multitasking on the computer, both the study and test phases were conducted in full-screen mode. Participants were informed that leaving full-screen mode could result in no payment. This was made possible by Gorilla (the platform in which we programmed the experiment), which recorded the browser and screen sizes. At the end of the study phase, participants were prompted to explain how they learned the words in a sentence. Our intention was to exclude participants who mentioned writing down or recording the words, though none reported having done so.

## 2.7 POSITIVE CONTROLS

Given the online nature of our experiment, we first assessed whether our data exhibit anticipated behavioural effects. Specifically, we examined the presence of the DRM false memory effect, the serial position effect, and a potential sleep benefit in correct recall. This scrutiny is essential to establish the validity and robustness of our recall data.

### 2.7.1 DRM false memory effect

On average, participants recalled 2.58 (or 12.9%) critical lures. Given free recall, the chance level of a lure being produced is 0. We ran two one-sample $t$-tests, one within the Immediate groups, the other within the Delay groups. They showed that both groups were susceptible to false recall [Immediate: $t(247) = 20.90$, $p < .001$; Delay: $t(239) = 17.59$, $p < .001$], demonstrating the classic DRM false memory effect.

### 2.7.2 Serial position effect

It is expected that the first (1st) and last (8th) words in our DRM wordlists would be better recalled than those in the middle due to primacy and recency effects, respectively (e.g., Ebbinghaus, 1885; Mak et al., 2021; Nipher, 1878; Tse & Altarriba, 2007). This was indeed the case in our data, as both the Immediate and Delay groups exhibited the well-established U-shaped serial position curve (see Figure 2). Then, I used a Wilcoxon Signed Ranked Test to compare each participant's proportion of recall in the 1st position against that in the middle positions (i.e., the average in the 4th and 5th positions), confirming a primacy effect ($z = -17.09$, $p < .001$). I also ran the same test comparing a participant's recall proportion in the 8th position against that in the middle positions. It confirms a recency effect ($z = -4.87$, $p < .001$).

### 2.7.3 Sleep benefit on studied word recall

Decades of evidence suggests that newly encoded declarative memories are better remembered after sleep than after an equivalent amount of wakefulness (e.g., Plihal & Born, 1997). It is, therefore, expected that the Sleep group would recall more studied list words than their Wake counterparts (e.g., Lahl et al. 2008). This was indeed the case [$M_{Sleep} = 17.33$ (SD = 16.59) vs. $M_{Wake} = 13.41$ (SD = 12.93); see Figure 3], and an independent $t$-test showed that this was statistically significant [$t(237.63) = 2.20$, $p = .029$].[3]

I also performed an additional independent $t$-test, where the dependent variable was "adjusted correct recall", which subtracts the number of intrusions (i.e., neither the studied list words nor the lures) from the number of correct recalls (Diekelmann et al., 2010; Payne et al., 2009). In line with the previous $t$-test, it also revealed superior performance in the Sleep (vs. Wake) participants [$M_{Sleep} = 10.55$ (SD = 19.66) vs. $M_{Wake} = 3.31$

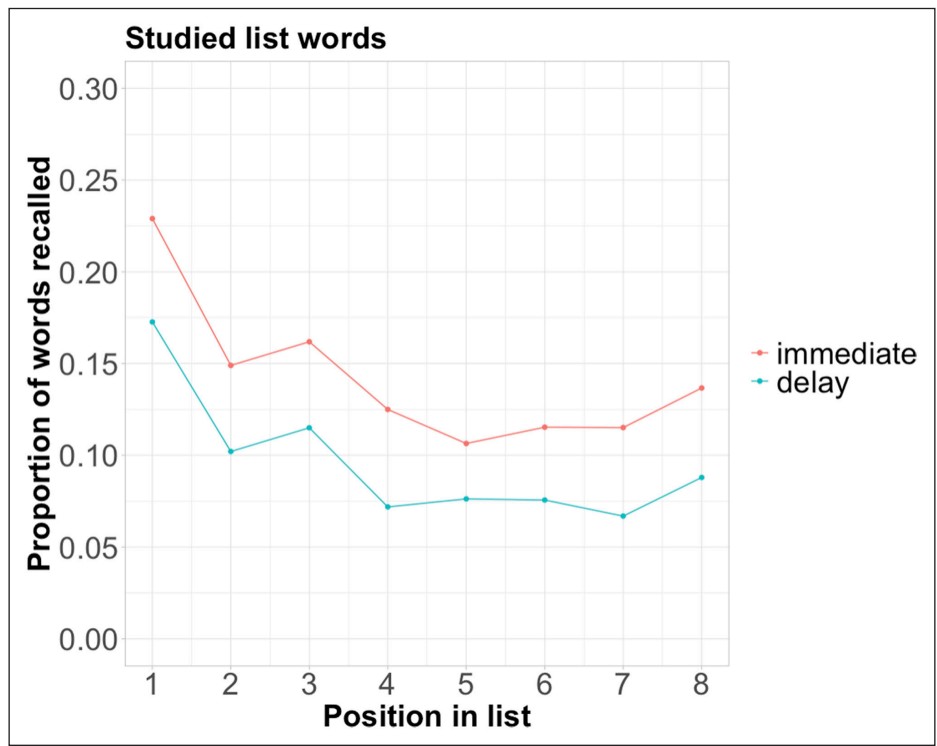

**Figure 2** Mean proportion of words recalled in each of the serial positions, summarised across the Immediate and Delay groups.

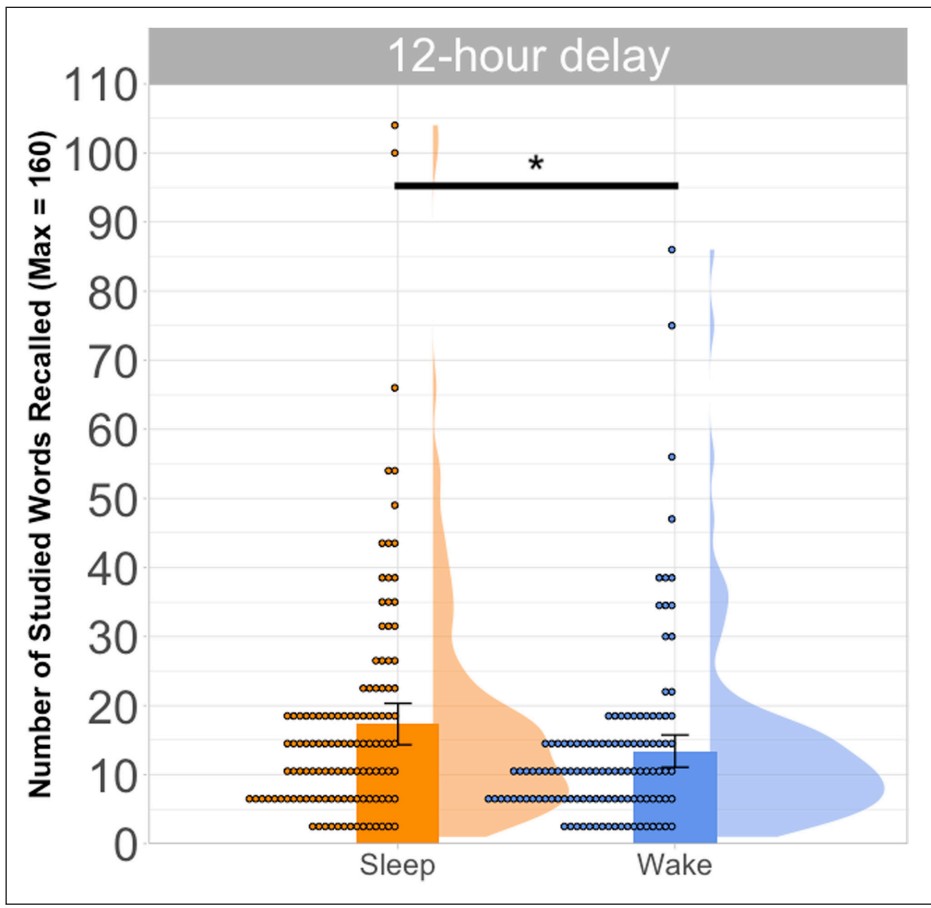

**Figure 3** Number of correct recalls across the Sleep and Wake groups. Each dot represents an individual participant and the error bars represent 95% confidence intervals.

(SD = 17.31); $t(234.26) = 3.03$, $p = .003$]. In short, our data demonstrated a clear sleep (vs. wake) benefit in studied word recall.

In summary, our recall data unveiled the classic DRM false memory effect, the serial position curve, and a sleep benefit in studied word recall. These observations underscore the strength and dependability of our online-collected data, attesting to its solid foundation for meaningful analyses.

## 2.8 MAIN RESULTS

The data reported here were collected to shed light on the effect of sleep in the DRM paradigm. Our registered report (Mak, O'Hagan, Horner, & Gaskell, 2023) described the findings in detail, so I will only a provide a brief summary of the key findings here.

1. The sleep group produced significantly fewer intrusions than the wake group.
2. When intrusions were statistically controlled for, there was evidence for greater false recall of the critical lures after sleep (vs. wakefulness). However, when intrusions were not controlled for, this sleep-wake difference became non-significant.
3. Regardless of whether intrusions were controlled for, there was clear evidence of the sleep group outperforming the wake group in veridical recall of the studied list words.
4. There were significant Interval × Test Time interactions in all the above analyses, indicating that these findings are above and beyond any time-of-day effects.

## 2.9 DATA ANONYMISATION AND ETHICAL ISSUES

This research received ethical clearance from the Department of Psychology Ethics Committee at the University of York. All participants provided informed consent before commencing the study. No personally identifiable data were collected unless participants in the delay groups voluntarily provided an email address to receive a reminder to take part in Session 2.

Each participant was associated with two distinct IDs: one was randomly generated by Gorilla (the platform in which we programmed the experiment), while the other was a participant-specific Prolific ID. While the former is untraceable to individuals, the latter holds some traceability. To maximise anonymity, Prolific IDs were removed from all the publicly shared datasets.

## 2.10 EXISTING USE OF DATA

Mak, M. H. C., O'Hagan, A., Horner, A. J., & Gaskell, M. G. (2023). A registered report testing the effect of sleep on Deese-Roediger-McDermott false memory: greater lure and veridical recall but fewer intrusions after sleep. *Royal Society Open Science, 10*(12). https://doi.org/10.1098/rsos.220595.

## 3. DATASET DESCRIPTION AND ACCESS

### 3.1 REPOSITORY LOCATION
The repository is located on Open Science Framework (doi: 10.17605/OSF.IO/P5JRH). URL Link to the repository: https://osf.io/p5jrh/.

### 3.2 OBJECT/FILE NAME
See Table 3 for details.

### 3.3 DATA TYPE
See Table 3 above.

### 3.4 FORMAT NAMES AND VERSIONS
All the data are available in comma-separated values (csv) format.

### 3.5 LANGUAGE
British English

### 3.6 LICENSE
CC0.

### 3.7 LIMITS TO SHARING
There is no limit to sharing.

### 3.8 PUBLICATION DATE
All the datasets were published in the OSF repository on 24th/25th April 2024.

### 3.9 FAIR DATA/CODEBOOK

1. Findability: To enhance findability, we have assigned descriptive metadata and tags to the recall data and repository.
2. Accessibility: We chose a reputable and widely recognised data repository (OSF) for hosting our datasets. Furthermore, we made our data available in csv to ensure compatibility with various data analysis tools.
3. Interoperability: To achieve interoperability, we have structured our dataset in a highly accessible and comprehensive way, enabling effective aggregation.
4. Reuse: We have taken proactive measures to facilitate dataset reuse. Comprehensive documentation, including a detailed README file, has been uploaded to OSF to help guide users through the recall data.

## 4. REUSE POTENTIAL

The dataset resulting from our DRM experiment holds substantial potential for reuse. I first highlight the strengths and limitations of our dataset, before suggesting how it can be reused.

| FOLDER NAMES | FILE NAMES | DESCRIPTION | DATA TYPE |
|---|---|---|---|
| 1. Study phase data | Order_of_Presentation_In_Study_Phase.csv | This spreadsheet shows the sequence of list presentation during the study phase for each participant. Since each participant saw 160 words presented one after the other and since there are 534 participants (before exclusion), this dataset has 160 × 534 observations (= 85440 observations). | Primary data |
| | DRMwordlists_used.csv | This shows the 20 DRM wordlists used in the experiment. | Material |
| 2. Recall data | full.csv | This is the raw free recall data, showing participants' responses exactly as they were put down, maintaining their original spelling and sequence. This dataset encompassed all 534 participants and has (160 studied list words + 20 lures) × 534 observations. | Primary data |
| | lure_final.csv | This is a simplified dataset, showing whether a critical lure was produced by a participant (Note that data from the 46 excluded participants were NOT included in this dataset, hence a sample size of 488). *Number of observations = 488 × 20 critical lures = 9760* | Processed data |
| | studied_final.csv | This is a simplified dataset, showing whether a studied list word was recalled by a participant (Note that data from the 46 excluded participants were NOT included in this dataset, hence a sample size of 488). *Number of observations = 488 × 160 studied words = 78080* | Processed data |
| 3. Survey data | demographic_survey.csv | This is the demographic survey (e.g., age, gender, ethnicity) that participants filled out in the study phase. It also contains sleepiness rating. | Primary data |
| | rMEQ_survey.csv | This contains all the data from the reduced morningness/evening Questionnaire (rMEQ; Adan & Almirall, 1991). This survey also asked participants to indicate how bright/noisy their immediate surrounding is, their sleep duration/quality the night before and whether they had a period of nap between sessions (Wake group only). | Primary data |
| 4. Miscellaneous | List_of_Excluded_ParticipantIDs.csv | This spreadsheet shows the IDs of the 46 participants who met our pre-registered exclusion criteria. | Processed data |
| | Encoding_Test_Time.csv | This spreadsheet shows the times at which a participant started the study and test phases. Given 534 participants (before exclusion), this dataset has 534 observations. | Primary data |

**Table 3** File descriptions.

## 4.1 STRENGTHS

First, our dataset encompasses data from 534 participants, making it a valuable resource for researchers seeking statistical power in their analyses. This large sample size increases the generalisability of findings and the likelihood of detecting subtle behavioural effects.

Second, we pre-registered our study, including the experimental procedures, exclusion criteria, pre-processing steps, and analysis plans. Our pre-registration underwent a thorough peer-review process to ensure methodological rigour and transparency, reinforcing the robust foundation upon which our study was conducted.

Third, unlike prior studies in the 'Sleep × DRM' literature, our sample was not restricted to undergraduate students. Instead, our diverse participant cohort encompassed a broader range of individuals, enhancing generalisability and ecological validity.

Finally, our dataset emerges as a pivotal benchmark in an era where online experiments are increasingly prominent in psychological research. Our study is poised to play a crucial role in shaping the trajectory of future memory/sleep studies conducted online. Researchers venturing into this area can look to our dataset as a reference point that embodies the standards to which they should aspire.

This benchmark not only underscores the viability of online research in capturing complex cognitive processes but also encapsulates the meticulous considerations required to ensure methodological robustness.

## 4.2 LIMITATIONS

Our experiment was conducted online, potentially introducing variations in participant engagement, distractions, and environmental conditions compared to in-person settings. For example, online experiments had no control over participants' immediate surroundings, potentially introducing uncontrolled variables such as noise and light that might influence behavioural performance. As a remedy, we asked participants to provide information on their surrounding environment in the test phase survey, but no formal analyses were conducted to investigate these factors. Notwithstanding, we still observed clear evidence for the classic DRM false memory effect, serial position effect, and a sleep benefit in word recall. These offer reassurance of the reliability of our data. In fact, administering the experiment online, while losing control over some extra-experimental variables, offers potential advantages in other respects. For example, it mirrors how most participants typically encode information on

a daily basis. This bolsters the ecological validity of our study, enabling a closer approximation of how memory processes take place in real life.

Another key limitation of our registered report is that we recruited 18-to-25-year-olds only.[4] Therefore, our data cannot inform false memory performance or sleep-related memory effects in different age groups (e.g., see Scullin & Bliwise, 2015 for a review).

Finally, while we randomly allocated participants to the four experimental groups, the sample might still have inherent biases due to self-selection. For example, participants with a morningness preference may be more likely to take part in the study if they were assigned to the Immediate-PM/Wake groups (e.g., Mak et al., 2023; Experiment 1). Fortunately, however, the four experimental groups were well-matched on sleepiness rating and morningness/eveningness preference (see Table 1), providing some reassurance that the effect of self-selection was minimal.

## 4.3 POTENTIAL REUSE SCENARIOS
### 4.3.1 Theoretical Advancements
Our dataset provides an opportunity for theory-building and theoretical refinement (see Mak, O'Hagan, Horner, & Gaskell, 2023 for an in-depth discussion). Using our extensive dataset, researchers can test and refine existing memory and sleep models. The Activation/ Monitoring Framework, while neutral on the influence of sleep, may be refined using our dataset. For instance, researchers can use our data to test whether the efficiency of spreading activation may differ between Sleep participants with different sleep duration. Another theory that may benefit from our datasets is the iOtA framework (Lewis & Durrant, 2011), which makes an explicit prediction that sleep would lead to an increase in DRM false recall but may have a limited effect on the studied list words. Inconsistent with this prediction, our data show that veridical recall may be influenced by a night's sleep to a greater extent than lure recall. Researchers can use our data to shed light on the *relative* effect of a night's sleep on gist abstraction and veridical memory consolidation (e.g., correlating veridical and false recall), thereby tightening the iOtA framework.

### 4.3.2 Modelling Time-related Memory Decay and Memory Search Processes
Our recall data offers a valuable resource for researchers interested in the intricate dynamics of memory decay and search processes. By examining how recall rates change as a function of time since encoding, researchers can construct and refine mathematical models that simulate memory degradation and memory search (e.g., Kahana, 2020; Reid & Jamieson, 2023). This can shed light on the temporal characteristics of memory decay, the influence of interference, and the cognitive processes and strategies underlying memory search. Our dataset

provides a strong foundation for validating and refining these models, potentially enhancing our understanding of how memory retrieval unfolds over time.

An exciting avenue for exploration, as suggested by an anonymous reviewer, is to leverage AI and large language models to extract insights from our datasets. By applying machine learning techniques such as classification or regression to our datasets, researchers could potentially develop machine-learning models to estimate individuals' memory outcomes based on various contextual factors, including experimental conditions, participant characteristics, and response patterns. This innovative approach holds promise for advancing our understanding of human memory and enhancing the capabilities of AI models in mimicking human cognition.

### 4.3.3 Present vs. Past data
The lexical items in the DRM paradigm have been extensively normed. The norming study by Stadler and colleagues (1999) remains arguably the most popular, despite being conducted over two decades ago. We used 20 of the 36 DRM lists from Stadler et al., although we shortened them to 8 words/list (from 15/list). As a matter of fact, false recall rates tend to be lower when fewer associates/list are used (Robinson & Roediger, 1997), so it is reasonable that our overall false recall rates are markedly lower than Stadler et al.'s (see last row of Table 4). However, fascinatingly, the relative rank of a lure's recall probability is noticeably different between Stadler et al. and ours. One example is the critical lure "window", which is the most frequently produced lure in Stadler et al. (i.e., 65% of their participants produced this lure), but in our dataset, it ranked at 7th (out of 20) in both our Immediate (14.5% of our participants produced this lure) and Delay (14.6% of our participants produced this lure) groups. Another example is "cold", which claimed the top spot in our data, but it did not even reach top 10 in Stadler et al., ranking at 15th (out of 20). These differences are intriguing and may be a consequence of various factors, including, but not limited to, number of associates per list, changes in associative strength over the years, different populations (American vs. British), and different experimental settings (lab vs. online; audio vs. visual exposure). These factors need to be examined further, but the difference in relative ranks between past and present data highlights the possibility of a potentially evolving landscape within the DRM paradigm. This investigation offers an opportunity to uncover subtle shifts and adaptations that memory mechanisms may undergo across time and contextual conditions.

### 4.3.4 Lexical Properties and Recall
Lexical properties, such as frequency, concreteness, and degree centrality play a role in memory processes (e.g., Brysbaert et al., 2018; Criss & Shiffrin, 2004; Madan, 2021; Mak & Twitchell, 2020; Mak et al., 2021). Researchers can leverage our recall data to unravel the interplay

| CRITICAL LURES | PERCENTAGE OF PARTICIPANTS WHO FALSELY RECALLED THE LURE (RELATIVE RANK WHERE 1 HAS THE HIGHEST PERCENTAGE AMONG THE 20 LURES) | | |
|---|---|---|---|
| | STADLER ET AL. (N = 205) | IMMEDIATE GROUPS (N = 248) | DELAY GROUPS (N = 240) |
| *Window* | **65% (1)** | **14.5% (7)** | **14.6% (7)** |
| *Sleep* | *61% (2)* | *16.9% (3)* | *15% (6)* |
| *Doctor* | *60% (3)* | *18.1% (2)* | *15.8% (4)* |
| *Smell* | **60% (3)** | **12.1% (11)** | **12.5% (9)** |
| *Chair* | *54% (5)* | *13.7% (8)* | *21.2% (2)* |
| *Smoke* | *54% (5)* | *16.5% (5)* | *15.8% (4)* |
| *Sweet* | **54% (5)** | **13.3% (10)** | **11.7% (10)** |
| *Rough* | *53% (8)* | *12.1% (11)* | *10.4% (13)* |
| *Needle* | *52% (9)* | *14.9% (6)* | *14.6% (7)* |
| *Anger* | *49% (10)* | *13.7% (8)* | *10% (14)* |
| *Rubbish* | *49% (10)* This is the percentage for 'trash' | *5.6% (19)* | *7.1% (20)* |
| *City* | *46% (12)* | *9.7% (14)* | *9.6% (16)* |
| *Soft* | *46% (12)* | *8.9% (15)* | *10% (14)* |
| *Cup* | *45% (14)* | *7.3% (18)* | *8.3% (18)* |
| *Cold* | ***44% (15)*** | ***29.4% (1)*** | ***25.8% (1)*** |
| *Mountain* | *42% (16)* | *10.1% (13)* | *11.7% (10)* |
| *River* | ***42% (16)*** | ***16.9% (3)*** | ***19.6% (3)*** |
| *Slow* | *42% (16)* | *5.2% (20)* | *8.8% (17)* |
| *Spider* | *37% (19)* | *7.7% (17)* | *10.8% (12)* |
| *Foot* | *35% (20)* | *8.5% (16)* | *8.3% (18)* |
| Mean | 49.5% | 12.76% | 13.08% |

**Table 4** Percentage of participants who falsely recalled a critical lure and the relative ranks of each lure in Stadler et al (1999) and our data.

*Note.* (1) A percentage of 50% means that half of the sample size falsely recalled that lure. (2) The bold items are those whose relative ranks differed by >= 5 between Stadler et al. and our Immediate groups.

between these lexical attributes and recall rates. This exploration could reveal insights into the mechanisms of word encoding, organisation, and retrieval. Additionally, researchers may want to test whether susceptibility to false recall is related to various lexical properties (e.g., Cann et al., 2011), further deepening our understanding of the factors influencing memory distortion. Finally, the published dataset, full.csv, also contains the raw, non-processed responses provided by participants, so researchers can potentially make use of such data to investigate e.g., how lexical properties may relate to typos/spelling mistakes. It is also possible to use these

raw data to investigate if there is e.g., a common trend in the semantic properties of the intrusions (i.e., responses that were neither the studied words nor the critical lures).

### 4.3.5 Individual Differences

We collected information such as gender, sleep duration, education attainment, and morningness/eveningness preference. This provides a fertile ground for investigating the realm of individual differences. Researchers can scrutinise how these factors contribute to variations in both veridical and false recall (see Kuula et al., 2019 for an example). This exploration can unravel the complex web of cognitive, demographic, and experimental factors that influence memory functions. By embarking on these avenues of exploration, researchers can contribute to the broader discourse on memory functioning while harnessing the full potential of our large and comprehensive dataset.

## 5. CONCLUSION

In summary, our rich DRM dataset possesses considerable potential for reuse across various research domains. Its extensive sample size, methodological transparency, and detailed demographic information offer a rich platform for advancing theoretical, practical, and interdisciplinary inquiries. Our dataset serves as a valuable resource with enduring impact beyond its original study objectives.

## APPENDIX A – EXPRESSION OF INTEREST SURVEY

Page 1: Demographic information

1) What is your gender identity?
2) How old are you?
3) In what country do you currently live?
4) What is your first language(s)?
5) What is your ethnicity?
6) What is the highest level of education you have completed?
7) Do you have any history of any psychiatric (e.g. schizophrenia), developmental (e.g. autism, dyslexia) or sleep disorders (e.g. insomnia)?
8) If your answer to the above is Yes, please can you name the diagnosis/es?

Page 2: Outline of the main study
IMPORTANT: Please read carefully We are recruiting hundreds of participants for a simple memory study. We would like to see if you may be interested in taking part.

In Task 1, participants will see and remember some English words. This will take about 10 min. In Task 2, participants will complete a simple memory test based on the words they saw in Task 1. This requires about 12 min.

Participants will receive £3.5 (£9.5/hour) upon completion of the two tasks. Importantly, participants will be randomly allocated to one of the four groups:

Group A (AM Group): You can start Task 1 and 2 any time between 8:30 and 10.30.
Group B (PM Group): You can start Task 1 and 2 any time between 8.30 and 22.30.
Group C (Delay Group 1): You can start Task 1 any time between 8.30–10.30 and then Task 2 between 8.30–22.30 on the same day.
Group D (Delay Group 2): You can start Session 1 any time between 8.30–22.30 and then Task 2 between 8.30–10.30 the day after.

Those in Groups C and D will receive a £0.2 bonus upon completion of the study. Unfortunately, we are NOT able to accommodate any preferences for group allocation.

If you are happy to take part in our memory study, press Yes below. If not, press No.

**Yes / No**

## NOTES

1  We planned to ask only the Sleep group to report their sleep duration and quality. However, due to a programming error, participants in the AM- and PM-control groups were also asked to indicate their sleep duration. These three groups totalled to 391 participants (before exclusion), and therefore, there were 391 observations for both the SleepTime and WakeTime variables in the rMEQ survey. For sleep quality, only participants in the Sleep group (N = 134; before exclusion) were asked to report that, and hence, 134 observations.

2  Because there is no point in asking participants in the control groups (which had no delay between study and test) or the sleep group (which would have slept) whether they had napped between study and test.

3  The number of correct recalls was log-transformed to give a more normal distribution.

4  This was intended to ensure comparability with previous 'Sleep × DRM' studies (e.g., Payne et al., 2009).

## ACKNOWLEDGEMENTS

I thank Dr Aidan Horner and Professor Gareth Gaskell (University of York) for their advice and mentorship throughout the project. I would also like to thank the two anonymous reviewers for their excellent suggestions.

## FUNDING INFORMATION

This project was funded by a British Academy/Leverhulme Small Research Grant (SRG21\210150), which was awarded to Matthew HC Mak and Gareth Gaskell in November 2021. The funders have no role in study design, data collection and analysis, decision to publish or preparation of the manuscript.

## COMPETING INTERESTS

The author has no competing interests to declare.

## AUTHOR CONTRIBUTIONS

Matthew H.C. Mak wrote the manuscript. The data were collected by Ms Alice O' Hagan (University of Oxford).

## AUTHOR AFFILIATIONS

**Matthew H. C. Mak** ⓘD orcid.org/0000-0001-7237-4931
University of Warwick, United Kingdom

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

## PEER REVIEW COMMENTS

*Journal of Open Psychology Data* has blind peer review, which is unblinded upon article acceptance. The editorial history of this article can be downloaded here:

- **PR File 1.** Peer Review History. DOI: https://doi.org/10.53 34/jopd.98.pr1

**TO CITE THIS ARTICLE:**

Mak, M. H. C. (2024). Data from "A Registered Report Testing the Effect of Sleep on DRM False Memory: Greater Lure and Veridical Recall but Fewer Intrusions After Sleep". *Journal of Open Psychology Data,* 12: 6, pp. 1–15. DOI: https://doi.org/10.5334/jopd.98

**Submitted:** 14 August 2023    **Accepted:** 12 May 2024    **Published:** 09 July 2024

