## [Peer Review History. · Journal of Open Psychology Data]

26th April, 2024

Dear Dr Evans,

I extend my sincere appreciation for your diligent efforts in finding reviewers. I am also grateful to the reviewers for their detailed and constructive comments on the manuscript, **Data from “Sleep, relative to wake, increases both veridical and false memory in the DRM paradigm: A registered report”**. The registered report, which has already been published, was re-named following peer review, so this submission has been renamed accordingly. The new title is **Data from “A registered report testing the effect of sleep on DRM false memory: Greater lure and veridical recall but fewer intrusions after sleep”**.

Both reviewers gave excellent suggestions on how to make the data more accessible. I have reformatted the data accordingly. However, since the data are associated with a published registered report, the reformatting (e.g., relabelling and removing some columns) would render some of the R scripts for the registered report un-usable. Replacing the original datasets with the reformatted ones will also remove the original timestamps, which I would rather avoid, as modifying the datasets *after* the registered report was published may raise some eyebrows. I have therefore created a new OSF repository for this JoOPD submission so that it contains all the reformatted datasets, organised in an easily accessible format. Below, I provide a point-by-point response explaining how each comment is addressed. Please note that in the revised manuscript, I have used tracked changes; in this response letter, my responses to reviewers are highlighted in **blue**, while paragraphs in **orange font colour** represent new additions to the manuscript.

I look forward to the outcome of your assessment. Thank you very much!

With best wishes,
Dr Matthew Mak

Assistant Professor
Department of Psychology,
University of Warwick,
Coventry, UK

All in all, the author provides a clean and high-quality dataset that has a great potential of reuse and, thus, provides a high value to sleep and memory researchers interested in false memory / gist abstraction, but also to the memory community more generally. I recommend publication after considering a few minor comments, as detailed below.

I thank the reviewer for their compliment and excellent suggestions.

Comments on the dataset:

1. I found it a bit confusing that some of the files (e.g., “full.csv”) contain empty columns (such as the columns “Local.Date”, “word”, “class”, etc.). If these columns are meaningful in context of the other variables in this file and cannot be found in any other file, they should be filled with the respective information. If they are not meaningful or “can be ignored” (as the author states in the corresponding “README.pdf” file), I recommend removing these columns altogether.

I apologise for the confusion. All the empty columns have now been removed from full.csv.

2. The “Time.csv” file appears to include only data from the delayed groups (Sleep and Wake) and should also include start times for the control groups (Morning and Evening).

These have now been added. The file has been renamed to Encoding_Test_Time.csv.

3. In the “lure_final.csv” and “studied_final.csv” files I found the name of column I (“answer”) misleading given that this column does not refer to the participants’ answer but rather to the “correct answer” / “correct response” or “lure word” (lure_final.csv) / “studied word” (“studied_final.csv”). I suggest renaming these columns accordingly. I also suggest removing column L (“intrusions”) in both files, given that this information is unrelated to lures and studied words. Also, I found it unclear why the same number is shown multiple times. I find that this information can be easily extracted from the “full.csv” file.

The column name ‘answer’ has now been replaced with ‘lure_word’ and ‘studied_word’ respectively. I have also removed the intrusion columns from both datasets.

4. The values in the “order” columns of both “lure_final.csv” and “studied_final.csv” files are not continuous in the way that they either don’t start with “1” or are missing some numbers inbetween. Are there missing entries? Please clarify / correct.

They are not missing entries. The lure_final.csv and studied_final.csv were extracted from full.csv, which contains both the lures and studied words. So when the lures were extracted, the studied words (and their accompanying row numbers) were discarded, creating an illusion that there are missing entries in the lure_final.csv. I have now updated the column, order, in both lure_final.csv and studied_final.csv so that it has no ‘missing’ numbers.

5. In the “second_survey.csv” file, please also add the final rMEQ scores, indicating participants’ chronotype.

The dataset has now been renamed to ‘rMEQ_survey.csv’, following the suggestion by the second reviewer. The final rMEQ scores for each participant have now been added to this file. To locate these scores, filter Question.Key = “Final_MEQ_Score”.

I also noticed that some of the variables in this file appear to not include an entry for all participants (e.g., the variable “NAP” only contains 143 entries, the variable “Sleep_quality” only contains 134 entries, the variable “sleepTime” only contains 391 variables and so on). Please clarify / correct.

The reasons why some of the variables in the file did not include an entry for all participants is because those questions were not asked in every group. For example, the NAP variable asked participants in the wake group to indicate whether they had a nap in the 12 hours between study and test. There is no reason to ask participants in the control groups the same question as there is no delay between study and test. There is also no reason to ask the Sleep group to report that because they would have reported their sleep/wake time. Since there are 143 participants in the Wake group (before exclusion), there are 143 observations for the NAP variable. I have made this point clear as a footnote (footnote 2) on p. 8 of the manuscript now.

For sleep duration/quality, we planned to ask only the Sleep group to report their sleep duration and quality. However, due to a programming error, participants in the AM- and PM-control groups were also asked to indicate their sleep duration. Since the Sleep, AM, and PM-Control groups totalled to 391 participants (before exclusion), there were 391 observations for both Sleep Time and Wake Time in the rMEQ survey. For sleep quality, only participants in the Sleep group (N = 134; before exclusion) were asked to report that, hence, 134 observations. I have now added this point as a footnote (footnote 1) on p. 8 of the manuscript now.

6. The author should consider including an information on the sequence of list presentation during the study phase for each participant.

This is a good idea. I have created a spreadsheet showing the sequence of list presentation for each participant (“Order_of_Presentation_In_Study_Phase.csv”). Since there were 534 participants (before exclusion) and each was presented with 160 words, there were $534 \times 160 = 85440$ observations in this spreadsheet. The column, Trial Number (1-160), shows the exact order in which the 160 words were presented. The column, “word”, shows the studied list words presented. A README has also been added to help readers understand this spreadsheet.

7. In the “README.pdf”, “the first response but down by a participant” should probably read “the first response put down by a participant”?

Fixed, thank you.

Comments on the manuscript:

1. The study phase description in section 2.5.1 does not appear to match with the illustration in Figure 1 regarding the sequence of the language/attention check and the SSS. Which was done first? Please clarify / correct.

I apologise for the confusion. The figure is correct, SSS was indeed followed by a language/attention check. This order has now been corrected in the manuscript (p. 8):

Participants first gave informed consent, rated their level of sleepiness on the Stanford Sleepiness Scale (SSS; Hoddes et al., 1973), completed a language/attention check (detailed in the next section), filled in a demographic survey, and studied 20 DRM wordlists.

Also, I was a bit confused about the meaning of “survey” in Figure 1. Does it refer to the “second part of the survey” only? Please clarify and consider renaming the two survey parts to avoid confusion.

Figure 1 has now been updated. There were two surveys in the experiment, the first one being a demographic survey, administered in the study phase. The second one is a rMEQ survey which also asked participants to report e.g., level of brightness/noise in the immediate surrounding, whether they had a nap between sessions (Wake group only), etc. As per the request of the second reviewer, the surveys have now been renamed to “demographic” and “rMEQ” respectively, so the new Figure 1 should clear up any confusion.

Finally, in Figure 1 only one “attention check” is illustrated, although attention checks were also performed during learning and testing. Please consider adding a figure caption describing the procedure in more detail.

In the updated Figure 1, I have added “(inc. attention checks)” below “View 20 DRM wordlists” and “Free Recall” to indicate that attention checks were performed during learning and testing.

2. In Table 2, I suggest to also insert “(Note 1)” after the changed critical lure word “Rubbish”.
Added.

3. In addition to the DRM false memory effect, the author should consider also adding results on the effect of sleep on false memory formation as this was the main question of the study.

I have now added a brief section summarising the results on p. 11-12 of the manuscript:

2.8 Main results

The data reported here were collected to shed light on the effect of sleep in the DRM paradigm. Our registered report (Mak, O’Hagan, Horner, & Gaskell, 2023) described the findings in detail, so I will only provide a brief summary of the key findings here.

1. The sleep group produced significantly fewer intrusions than the wake group.
2. When intrusions were statistically controlled for, there was evidence for greater false recall of the critical lures after sleep (vs. wakefulness). However, when intrusions were not controlled for, this sleep-wake difference became non-significant.
3. Regardless of whether intrusions were controlled for, there was clear evidence of the sleep group outperforming the wake group in veridical recall of the studied list words.

4. There were significant Interval x Test Time interactions in all the above analyses, indicating that these findings are above and beyond any time-of-day effects.

4. The serial position effect was only described qualitatively. Consider also providing statistical evidence for this effect.

I have followed the reviewer's suggestion and added this to the manuscript (p. 10):

Then, I used a Wilcoxon Signed Ranked Test to compare each participant's proportion of recall in the 1st position against that in the middle positions (i.e., the average in the 4th and 5th positions), confirming a primacy effect ($z = -17.09, p < .001$). I also ran the same test comparing a participant's recall proportion in the 8th position against that in the middle positions. It confirms a recency effect ($z = -4.87, p < .001$).

5. In Figure 2, the y-axis description "Mean recall rates" appears incorrect. I assume that the values in this figure represent numbers of correctly recalled studied words rather than recall rates? Please clarify / correct.

The Figure has been updated and is now showing the proportion of words recalled in each serial position (see p. 10).

6. In section 2.8 ("Existing use of data"), the "in-principle acceptance" should probably be updated given that the article has already been published (Mak et al. 2023, R. Soc. Open Sci.)?

This has been fixed, thank you!

7. In section 3.1, please consider, in addition to indicating the DOI, to also provide a direct link to the data repository for quick and easy access.

This has now been added.

Reviewer d:

Comments to the author(s)

The authors provide details for a dataset from a registered report testing the role of sleep on false memory using a DRM paradigm with lists of semantically related words. The dataset was collected online, and is quite large, with young, healthy participants, who completed either short or delayed recall, the delay condition included participants who slept and those who stayed awake during the 12-hour retention interval. The authors were able to replicate typical, expected effects, and offer suggestions for how these data could be used by future researchers.

Below I provide some feedback about where to clarify details of the methods section, and reuse of the data. I also have a few comments on the organization of the data.

I thank the reviewer for the taking time to review this article and their excellent suggestions.

THE METHOD SECTION:

1. Recruitment/inclusion was done in stages. Stage one involved telling participants if they wanted to participate, they would be assigned to 1 of 4 groups (section 2.4 top of page 5). I think more detail here about what participants were specifically told about the study at this stage would help identify the characteristics of the sample and who chose to continue to the next phase.

The ‘expression of interest’ survey is now available in appendix A for reference (p. 23).

2. It isn’t clear how participants were contacted the following day for the delay group. Please provide the instructions to participants about when they needed to log back in to do the delayed test.

The relevant details have now been added to the manuscript (p. 8):

For participants in the sleep and wake groups, Session 1 ended after wordlist exposure. On the final page, they were told that they should take part in the second session, which would be available 12 hours later, accessible via Prolific. They were also encouraged to put down an email address if they wished to be reminded via email. We programmed the experiment such that participants who put down an email address received an automated email when the second session became available 12 hours later. Over 75% of the participants chose to supply an email address.

3. Table 1 could include more detail about the sample race and ethnicity rather than just reporting the percentage of white participants. This would help determine whether this is a variable that could be used in future use of the data.

We provided 6 options for participants to choose from: Asian, Black/Caribbean, Latino, Mixed, Other, and White. The distribution in each group has now been added to Table 1 (p. 6).

4. Section 2.5.1 “This specific test format wasn’t specified”. Please clarify what you mean by this.

This has now been made clearer in the manuscript (p. 8):

Participants were told that their “memory for the words will be tested later”, but there was no mention of the specific test format (i.e., participants were not told that they would engage in free recall).

5. One of the attention checks involved the auditory presentation of a story, but it wasn’t noted whether participants knew they would need audio or if this was an exclusion criterion if their computer did not have audio/they were in a space that did not allow audio.

This section has now been clarified (p. 9):

Afterwards, participants were told to turn on their audio so they could listen to a short story in English (14 sec). Participants heard the story once, without the option to replay or pause. Participants then responded to two simple comprehension questions. These served as language/attention checks, ensuring that participants understood English and were in a reasonably quiet place. Three individuals did not get both questions right, so the study terminated there for these individuals. Since they could not complete the study/test phase, they did not meet our inclusion criteria (and therefore not counted towards the sample size).

6. Table 4 is a bit difficult to interpret because of the different number of participants in each column. It might be more intuitive and easier to make meaningful comparisons if percentages were reported rather than the raw number of participants.

Each cell in Table 4 shows the percentage of participants who falsely recalled a critical lure, not the raw number of participants. I have added % to each cell to make this clearer and changed the table title from “false recall rates..” to “Percentage of participants who falsely recalled a critical lure...”. I have added the following to help orient readers (p. 16):

However, fascinatingly, the relative rank of a lure’s recall probability is noticeably different between Stadler et al. and ours. One example is the critical lure “window”, which is the most frequently produced lure in Stadler et al. (i.e., 65% of their participants produced this lure), but in our dataset, it ranked at 7th (out of 20) in both our Immediate (14.5% of our participants produced this lure) and Delay (14.6% of our participants produced this lure) groups. Another example is “cold”, which claimed the top spot in our data, but it did not even reach top 10 in Stadler et al., ranking at 15th (out of 20).

Note for Table 4 (p. 17)

(1) A percentage of 50% means that half of the sample size falsely recalled that lure.

7. Tiny detail, but the description in section 2.5.1 indicates language/attention checks, and then SSS was done, but Figure 1 has the reverse order. Please clarify the order.

Apologies for the confusion. The figure is correct, but the verbal description in section 2.5.1 was not. The relevant section has now been fixed (p. 8).

8. Please indicate whether all participants, even those who participated in 2 sessions, were compensated the same amount.

This information has now been added to the manuscript (p. 5):

All participants were reimbursed at a rate of £9.5/hr, but those in the Delay groups received a bonus of £0.2 after completion of both sessions.

THE REUSE SECTION:

1. One of the noted suggestions for potential reuse is refinement and validation of DRM-related theories such as Fuzzy-Trace Theory and Activation/Monitoring Frameworks, but these ideas are not fleshed out so it is difficult to determine how useful these data would be to examine this.

I was hesitant to go into the theoretical side of the data, because this paper is meant to provide a relatively *atheoretical* description of the dataset. Our registered report has an extensive discussion of how our data may inform theoretical development, and readers are encouraged to refer to that paper if they are interested. Having said that, I do see some value in a very brief discussion of theories, so I have added the following to the manuscript (p. 15):

Our dataset provides an opportunity for theory-building and theoretical refinement (see Mak, O'Hagan, Horner, & Gaskell, 2023 for an in-depth discussion). Using our extensive dataset, researchers can test and refine existing memory and sleep models. The Activation/Monitoring Framework, while neutral on the influence of sleep, may be refined using our dataset. For instance, researchers can use our data to test whether the efficiency of spreading activation may differ between Sleep participants with different sleep duration. Another theory that may benefit from our datasets is the iOtA framework (Lewis & Durrant, 2011), which makes an explicit prediction that sleep would lead to an increase in DRM false recall but may have a limited effect on the studied list words. Inconsistent with this prediction, our data show that veridical recall may be influenced by a night's sleep to a greater extent than lure recall. Researchers can use our data to shed light on the relative effect of a night's sleep on gist abstraction and veridical memory consolidation (e.g., correlating veridical and false recall), thereby tightening the iOtA framework.

2. Another possible addition to section 4.3.4 would be to provide the raw, non-pre-processed data in case there are future research questions about common typos/inflections, and additional false memories that are not the critical lure.

This is a good idea. The spreadsheet ‘full.csv’ contains all the raw, non-processed data so researchers can indeed readily use that dataset to investigate the questions proposed by the reviewer. I have added the following to section 4.3.4 (p. 17-18):

Finally, the published dataset, full.csv, contains the raw, non-processed responses provided by participants, so researchers can potentially make use of such data to investigate e.g., how lexical properties may relate to typos/spelling mistakes. It is also possible to use these raw data to investigate if there is e.g., a common trend in the semantic properties of the intrusions (i.e., responses that were neither the studied words nor the critical lures).

3. I wonder if the authors have thought about AI/large language models as a way to reuse the data. I don’t have any specific suggestions, that’s not my area of expertise but it came to mind that there are perhaps ways to leverage this technology to pull out other types of patterns.

This is a very interesting idea, and I agree that applying AI/LLM to the dataset can be a potential avenue for future research. I have added this suggestion to the manuscript now (p. 16).

An exciting avenue for exploration, as suggested by an anonymous reviewer, is to leverage AI and large language models to extract insights from our datasets. By applying machine learning techniques such as classification or regression to our datasets, researchers could potentially develop machine-learning models to estimate individuals’ memory outcomes based on various contextual factors, including experimental conditions, participant characteristics, and response patterns. This innovative approach holds promise for advancing our understanding of human memory and enhancing the capabilities of AI models in mimicking human cognition.

THE DEPOSITED DATA

1. I would reconsider the file name “first_survey.csv” This is not very descriptive, but could be called demographics or sample characteristics.

This has now been renamed to “demographic_survey.csv”.

2. I had a hard time finding the data on the OSF page because there are different studies and multiple folders. Including file paths in Table 3 and maybe in the descriptions on OSF would be helpful.

The datasets have now been moved to a new OSF repository, and this repository is organised in a significantly more accessible manner. I have also added a column to Table 3 in the manuscript, which shows the folder name where each dataset is located.

3. section 3.9 findability – I didn’t see any assigned descriptive metadata under the “metadata” tab on OSF.

This has now been added, thank you.